# Influence of Materials on Dry Friction and Wear Performance of Harmonic Reducer Circular Spline

**Guyue Hu [1], Yi Ge [2], Tong Wu [3], Xiaobiao Mu [4], Fengyao Ren [3], Zhuhao Shao [3] and Chaolei Zhang [2,\***

[1] School of Civil and Resource Engineering, University of Science and Technology Beijing, Beijing 100083, China
[2] Innovation Research Institute for Carbon Neutrality, University of Science and Technology Beijing, Beijing 100083, China
[3] School of Materials Science and Engineering, University of Science and Technology Beijing, Beijing 100083, China
[4] Beijing CTKM Harmonic Drive Co., Ltd., Beijing 101318, China
[\*] Correspondence: zhangchaolei@ustb.edu.cn

**Abstract:** Ductile iron and alloy steel are mostly used as the circular spline materials of harmonic reducers. The study of the differences in wear resistance of different circular spline materials is a strong basis for the design of high-precision and long-life harmonic reducers. The dry friction behaviors under room temperature of two ductile iron (low-load SHF and high-load SHG) alloys and steel (40Cr) circular spline materials were studied with a quantitative analysis of the alloy composition, microstructure, hardness and wear resistance. The results showed that the microstructures of SHF, SHG and 40Cr were graphite nodules within a tempered sorbite matrix, graphite nodules within a pearlite matrix and tempered sorbite, respectively. The hardness was inversely proportional to the wear resistance. The wear resistance of ductile iron was better than that of 40Cr, with SHF having the best wear resistance. The microstructure of the SHF and SHG ductile iron had a length of 8.1 μm, 5.9 μm (Ti, V, Mo)/C and Cu/Mg second-phase particles. Compared with tempered sorbite, the self-lubricating layer formed by the graphite nodules can effectively reduce wear. The proportion and average diameter of the SHF graphite were 1.2% and 30.9% higher, respectively, than those of the SHG graphite, providing a higher graphite spalling probability and longer graphite lubrication distance.

**Keywords:** harmonic reducer; ductile iron; 40Cr; dry friction; wear resistance



## 1. Introduction

The manufacturing industry has moved towards full automation with the trend of developing an "Industry 4.0 era". The harmonic reducer is widely used in industrial manufacturing, aerospace, national defense and other fields due to its compact size, light weight, high reduction ratio and accuracy [1–3], especially to promote the development of compact electromechanical integration of industrial robots [4]. According to the IFR World Robotics Report 2020, robot sales data show Europe's inventory growth in 2019 was 580,000 units, which was up 7%. Germany was still the main user; their operating stock was approximately 221,500 units, which was approximately three times that of Italy (74,400 units), five times that of France (42,000 units) and ten times that of Britain (21,700 units) [5]. Northeast Asian countries (Korea, Japan) and Singapore are the world leaders in adopting industrial robots, and the robot density of these countries is 4–6 times higher than the world average number of 69 units per 10,000 employees [6]. The technical barrier of the harmonic reducer is high. Worldwide, the HarmonicDrive ®(HD) is in a monopoly position in the industry, occupying approximately 70% of the market share of harmonic reducers [7]. Key technical issues, such as the tooth profile design, special materials, processing technology, testing and evaluation, must be broken through [8,9] in the process of research and development of the harmonic reducer. The harmonic reducer is composed of a flexspline, circular spline and wave generator. The wave generator transmits

power to the driven flexspline, and then the gear meshing transmission formed by the flexspline and circular spline continuously outputs power to the outside [10].

For a long time, studies on the harmonic reducer have focused on the finite element model analysis of the stress and strain on the flexspline and the wear failure analysis of the tooth meshing between the flexspline and the flexible bearing to improve the fatigue strength and wear resistance of the flexspline and the flexible bearing. Zhu Caizhi [11] studied the theoretical model of contact force between the rolling elements and outer ring based on three bending moment equations, and the characteristics of a CSF-40-80 flexible bearing were also studied with finite element method analysis; stress, deformation and velocity were established as linkages between the static and dynamic states. Bikash Routh [12] studied the geometry of the coning gap and its variation with the strain wave generating cam rotation, estimating the coning gap and lubrication pressure profiles. Vineet Sahoo [13] analyzed the stresses in the flexspline/gear cup in harmonic drives with involute toothed gear pairs and conventional strain wave generating cam with the finite element method in ANSYS and verified that the full load was distributed over all the possible primary and secondary contacts in proportion to their contact intensities. Congbin Yang [14] focused on a harmonic drive with a double circular-arc tooth profile and selected a suitable material and wall thickness to reduce the effect of the partial axial load of the flexspline; both the tooth width and chamfering of the flexspline teeth helped reduce the partial axial load and increase the flexspline length. To improve the flexspline tribological properties, the flexspline material 40Cr was modified with a robust polydimethylsiloxane (PDMS) coating [15]. Etched and chemically modified films were utilized to enhance the organic PDMS coating−substrate link strength. Comparing the modified and unmodified 40Cr, the surface friction coefficient decreased by 82.2%. To overcome the deformation difference between the inner and outer surfaces of the FS subjected to thermal load and force load during operation, Yangfan Li [16] analyzed the thermal−mechanical coupling deformation mechanism of the FS, and the structural parameters of the wave generator were optimized to eliminate the actual backlash and improve the actual transmission accuracy. It was reported that the wear mechanism of a flexspline with grease lubrication can be regulated with a novel amorphous/crystalline oxide form evolution at the frictional interface; the oxide layers consisting of sufficient amorphous FeOOH around $Fe_2O_3$ grains were dense, tough and favored anti-wear properties, whereas the oxide layers mainly consisting of crystalline $Fe_2O_3$ were unfavorable [9]. It was also confirmed that the surface treatment was beneficial to improving the wear resistance of the material; it was found that the TD coating can considerably resist sliding wear, which brings about the least friction coefficient, material transfer and material removal. The tool surface with nitriding treatment has the highest adhesion tendency to the workpiece material [17], while it is neglecting the research on the friction and wear of the circular spline. The hardness of the circular spline is often 3~7HRC lower than that of the flexspline because the flexspline is the main transmission component. Higher and lower hardness are found to result in stripping and adhesive wear, respectively [18]. In harmonic drives, matching the hardness of the tribo-pairs of the circular spline and flexspline determines the useful life of the harmonic drive; thus, a critical value exists for the hardness ratio of the tribo-pair components [19]. In order to ensure its bearing capacity and wear resistance, the wear resistance of the circular spline is being confronted with this trial. The material selection of the circular spline is different, and the cost and wear resistance are considered comprehensively. The wear of the circular spline surface has a serious impact on the transmission accuracy and stability of the harmonic reducer.

Recently, the circular spline is mostly constructed of carbon steel on the commercial market for a low production cost, but due to its low content of alloy, its hardenability is general, and it cannot be applied to occasions with a high load and high wear resistance. 40Cr alloy steel and 2Cr13 stainless steel have been utilized under special working conditions [20]. Compared with common steel, ductile iron has excellent characteristics, including easy cutting, self-lubrication, wear reduction, low cost and a cushioning property. A.S.M.A Haseeb [21] compared the tribological behavior of ductile iron heat-treated

with two different procedures, viz., quenching and tempering and austempering to an identical matrix hardness of 445 KHN, and found that ductile iron exhibited better wear resistance than quenched and tempered ductile iron. Lifeng Tong [22] found that ductile iron wheel material had excellent thermal properties and a higher carbon content and, therefore, exhibited a lower wear rate, a smaller difference value of the friction coefficient and a smaller plastic deformation on the worn surface than those of carbon steel wheel material. Jia-Hu Ouyang [23] reported that, if tested parallel to the basal planes, graphite is soft and lubricates in normal air; however, it fails to lubricate in a vacuum or at high altitudes and wears rapidly. In fact, its coefficient of friction in a vacuum or dry nitrogen is typically ten times greater than in air, but the existing research lacks the detection and evaluation of the wear resistance of different materials of the circular spline. Due to the immature inoculation technology, ductile cast iron is not widely used in precision drive parts [24,25]. In order to eliminate the influence of the lubricant on the wear state, this paper selected different materials of the circular spline to analyze the dry friction and wear performance. This was studied to provide theoretical support for the improvement of the wear resistance of the circular spline and ensure the high reliability and long life of the application end of the harmonic reducer drive mechanism.

## 2. Materials and Methods

The experimental materials were the products of the circular spline in three types of materials. SHF-20-50, constructed of ductile iron, is a small reduction ratio, low-load harmonic reducer, and it was supplied after quenching and tempering, which is represented by SHF below. SHG-25-80, constructed of ductile iron, has a larger reduction ratio and higher-load than SHF-20-50, and it was supplied after normalizing, which is represented by SHG below. The two materials were purchased from Harmonic Drive Systems Incorporation (Harmonic Drive Systems Co., Ltd., Shanghai, China). 40Cr is an alloy steel, used as a contrast material, supplied by Beijing CTKM Harmonic Drive Co., Ltd., Beijing, China, and it was supplied after quenching and tempering. The alloy composition was determined with the HCS-140 high-frequency infrared carbon sulfur analyzer, as shown in Table 1. All specimens were ultrasonically cleaned prior to use to remove any oil from their surfaces.

**Table 1.** Alloy composition of the tested ductile iron and 40Cr steel/wt.%.

| Alloy Compositon | C | Si | Mn | P | S | Cr | Ni | Cu | Mo | Ti | Fe |
|---|---|---|---|---|---|---|---|---|---|---|---|
| SHF | 3.17 | 2.18 | 0.40 | 0.021 | 0.007 | 0.043 | 0.015 | 0.19 | 0.005 | 0.015 | Bal. |
| SHG | 3.44 | 2.00 | 0.41 | 0.019 | 0.006 | 0.02 | 0.011 | 2.63 | 0.002 | — | Bal. |
| 40Cr | 0.40 | 0.21 | 0.63 | 0.019 | 0.005 | 0.93 | 0.02 | — | — | — | Bal. |

The morphology of the uneroded graphite and the microstructure after erosion by 4% alcohol nitrate were observed with a Zeiss Axio Scope. A1 optical microscope (OM, Carl Zeiss AG, Oberkochen, Germany). The nodularity, average diameter and grade of the graphite nodules were calculated with Image Tool and Nano Measurer; the nodularity and graphite nodule size were defined according to the 'ASTM A247-17 Standard Test Method for Evaluating the Microstructure of Graphite in Iron Castings.' The ZEISS Gemini 500 scanning electron microscope (SEM, Carl Zeiss AG, Oberkochen, Germany) was used to observe the microstructures before and after wear, the transmission electron microscope (TEM) JEM2100 (JEOL Ltd., Tokyo, Japan) was used to observe the microstructures of the HD type before wear, and the Olympus LEXT OLS400 3D laser-confocal microscope (Olympus Co., Ltd., Hamburg, Germany) was used to observe the surface wear morphology. The Rockwell and Brinell hardness were measured with the WHR-80D Whole Rockwell hardness tester (Zwick Roell Group, Ulm, Germany) and NHB7000 Brinell hardness tester (Shenyang Tianxing Testing Instruments Co., Ltd, Shenyang, Liaoning). Prior to the sliding wear tests, all specimens were ground with various grades of SiC paper and then polished to a final finish with 6μm diamond pastes. The friction properties of the materials were measured

with the UMT-2 multi-function testing machine (Center for Tribology Inc., Campbell, CA, USA) at room temperature (approximately 296.55 K) with a humidity of 53% under dry sliding with a reciprocating ball-on-plane tribometer consisting of a AISI 52100 steel ball rubbing against a static test sample; the main parts for the test are shown in Figure 1.

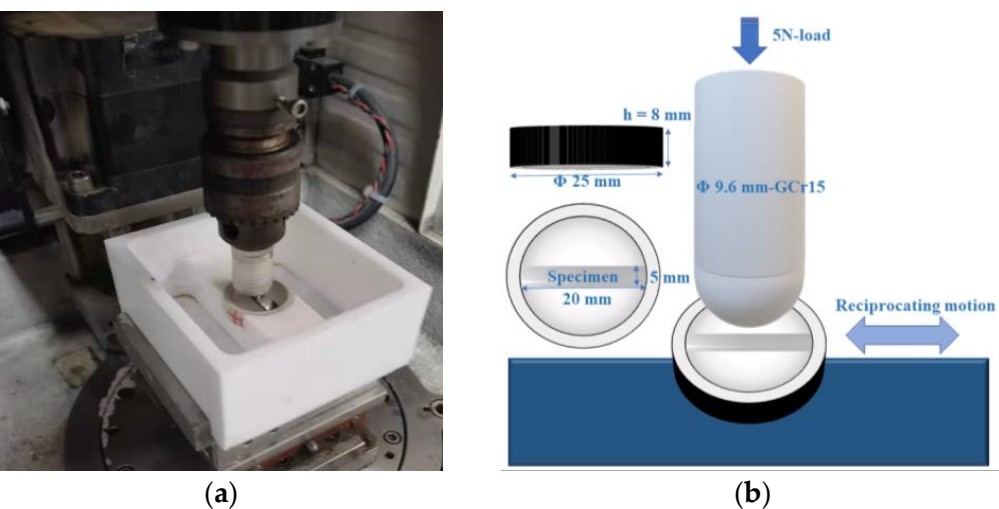

|        |        |
| :----: | :----: |
| (**a**) | (**b**) |

**Figure 1.** Experimental device: (**a**) physical diagram, (**b**) schematic diagram.

The sample was prepared by cold inlaying with epoxy resin and ethylene diamine (curing agent). The applied load was 5 N for studying the compact and low-load harmonic reducer circular spline, the frequency was 1 Hz, the linear velocity was 30 mm/s, and the experimental time was 1.5 h. The elastic modulus of ductile iron is 155 GPa, and Poisson's ratio is $\mu = 0.28$. The elastic modulus of 40Cr is 200 Gpa, and Poisson's ratio is $\mu = 0.3$. The elastic modulus of AISI 52100 is 208 GPa, and Poisson's ratio is $\mu = 0.30$. The radius of curvature of AISI 52100 is 4.8 mm, and the inverse radius of curvature of the tested piece is calculated as 0.

Considering that the contour of the contact surface is approximately a quadratic paraboloid, and the size of the contact surface is much smaller than the size of the object, the corresponding contact stress can be calculated with the Hertz formula. The stress applied to the sample surface was calculated according to the Hertz formula (1); the maximum contact stress was 749 MPa between ductile iron and AISI 52100, and the maximum contact stress was 813 MPa between 40Cr and AISI 52100. The average value of at least three repetitions was recognized as the final experimental result for each test.

$$\sigma_{Hmax} = \frac{1}{\pi} \sqrt[3]{6F_n \left( \frac{\frac{1}{\rho_1} + \frac{1}{\rho_2}}{\frac{1-\mu_1^2}{E_1} + \frac{1-\mu_2^2}{E_2}} \right)^2} \tag{1}$$

In the equation, $\sigma_H$: contact stress/MPa; $F_n$: Normal force/N; $\rho_1$, $\rho_2$: radius of curvature/mm; $E_1$, $E_2$: elastic modulus/MPa; $\mu_1$, $\mu_2$: Poisson's ratio.

## 3. Results

### 3.1. Microstructure and Morphology Analysis of the Circular Spline

Compared with alloy steel, ductile iron has graphite nodules in the microstructure, and the precipitation form and distribution of the graphite nodules have a significant impact on the hardness and wear resistance of the overall material [26]. It is necessary to compare and analyze the differences of the graphite nodules in the microstructures of the two circular splines and consider the internal reasons for the data difference and the influence on the external performance. Two types of graphite forms of the circular spline were observed after polishing, as shown in Figure 2a,b. SHF has more graphite nodules,

which are not as fully globalized as SHG, and the Ti content of SHF is 0.015%; however, the Ti content of SHG is negligible.

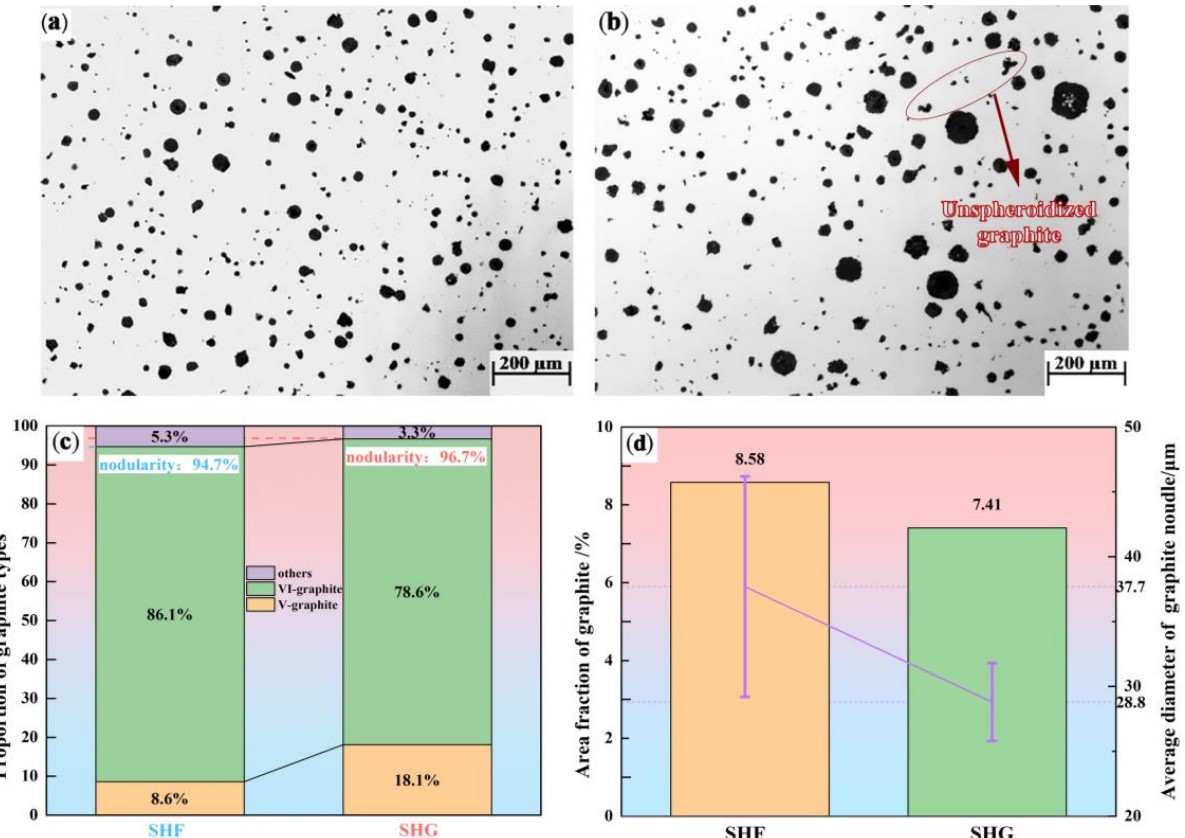

**Figure 2.** Morphology and count of the uneroded graphite: (**a**) SHF, (**b**) SHG, (**c**) nodularity of graphite, (**d**) area fraction and average diameter of graphite.

Ti is a spheroidal interfering element in ductile iron, which can cause the deformation of spherical graphite and affect the roundness of the graphite nodules [27]. According to the triple-repeated test results, the nodularity, area fraction and average diameter of the graphite were statistically analyzed, as shown in Figure 2c,d. The nodularity of the SHF graphite is 94.7%, and the nodularity of the SHG graphite is 96.7%. The roundness of both graphite nodules is great, higher than the high-quality ductile iron's 85% nodularity. The area fraction of the SHF graphite is 8.58%, while the area fraction of the SHG graphite is 7.41% with a difference of 1.17%. The size distribution of the SHF graphite nodules is uneven with an average diameter of 37.7 μm and a maximum diameter of 98.2 μm. The size of the SHG graphite nodules is more uniform with an average diameter of 28.8 μm. On the whole, the graphite area fraction and size of the SHF circular spline are larger than those of the SHG circular spline, and the nodularity is opposite to the uniformity. The content of Cu in the SHG circular spline reaches 2.63%, and the Cu eutectic shows negative segregation during solidification and is enriched in the eutectic cluster, which offsets part of the inhomogeneity of the microstructure caused by the positive segregation elements, such as Mn. In addition, it promotes graphitization during the eutectic transformation, does not form carbides with carbon, and promotes the generation of fine graphite nodules [28].

As shown in Figure 3, the matrix microstructures of the SHF and SHG circular splines are tempered sorbite and sheet pearlite, respectively, and the microstructure of the 40Cr circular spline is mainly tempered sorbite.. The pearlite lamellar spacing calculated with Image Tool was 0.57 μm.

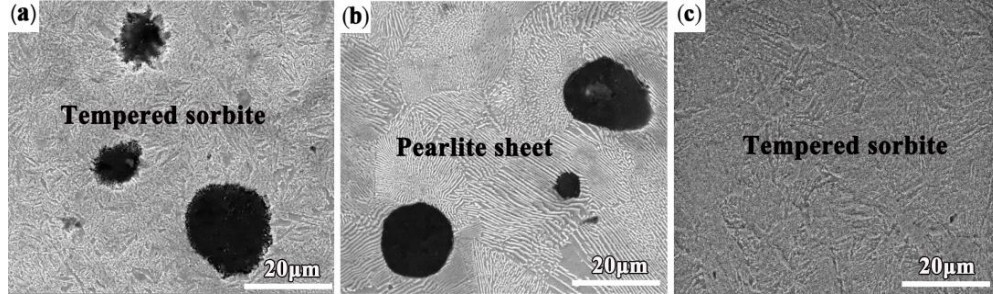

**Figure 3.** Microstructures of the circular splines: (**a**) SHF, (**b**) SHG, (**c**) 40Cr.

The morphology of the two ductile irons' precipitates was observed as shown in Figures 4 and 5, and both of the precipitates' average sizes are larger than 1 μm. Figure 4 shows that the large precipitates of the SHF circular spline are attached to oxides. The EDS shows that the precipitates on the tempered sorbite matrix are (Ti, V, Mo)/C with a maximum length of 8.1 μm and 3.5 μm, and there are also small (Ti, V, Mo)/C particles < 2 μm. Oxides provide particles for the nucleation of micro-alloyed carbides and thus preferentially grow there. When Ti content is low, TiC particles can be used as the heterogeneous core of the precipitated graphite, promoting graphite nucleation and increasing the count and area fraction of the graphite. At the same time, the increase in the graphite nucleation rate leads to the decrease in the supercooling degree of the solidification front and improves the nodularity [29]. Therefore, the overall excellent nodularity and area fraction of the SHF graphite are explained.

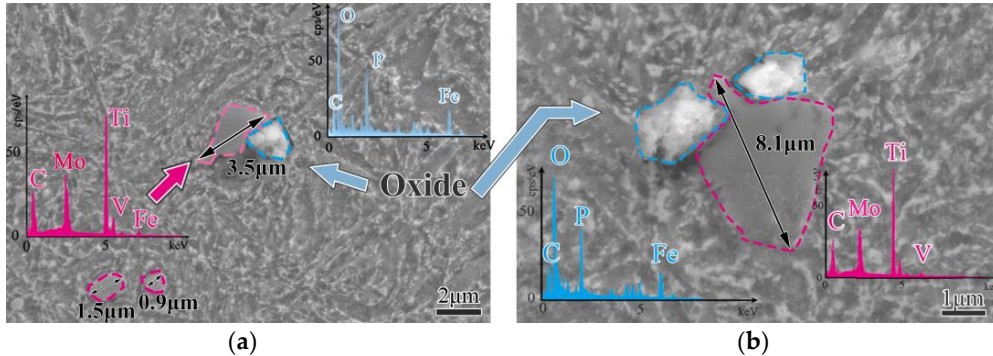

**Figure 4.** Morphology and EDS of the SHF precipitates: (**a**) precipitate 1, (**b**) precipitate 2.

The content of Cu in SHG is up to 2.63%, and Cu has a strong effect on promoting the precipitation of pearlite and refining the pearlite lamellar; however, at room temperature, the solubility of Cu in cast iron does not normally exceed 2%, and excessive Cu is considered to induce the brittleness of the material [30]. As shown in Figure 5a, the saturated spherical Cu/Mg second-phase particles have separated out along the grain boundaries with sizes of 5.9 μm and 3.6 μm. At the same time, long rod-like (Ti, V, Mo)/C precipitates with a length of 5.1 μm were precipitated within the grain. Micro-alloyed carbide can effectively pin austenite grain boundaries during high-temperature casting, disperse out and strengthen in the matrix during the low-temperature phase transition, refining the pearlite lamellar spacing and improving the strength and toughness of the matrix [31], as shown in Figure 5c.

The Cu-rich second phase, not larger than 100 nm, was diffused in the pearlite matrix. At a certain Cu content, the nucleation substrate of bovine ocular ferrite is destroyed, the diffusion of carbon into the graphite nodules is hindered, and the ferrite formation is hindered to basically obtain pearlite and improve the overall toughness [32]. At present, the Cu content of ductile iron is less than 2.0%, so it is necessary to further study the influence of high Cu content on the microstructure and properties.

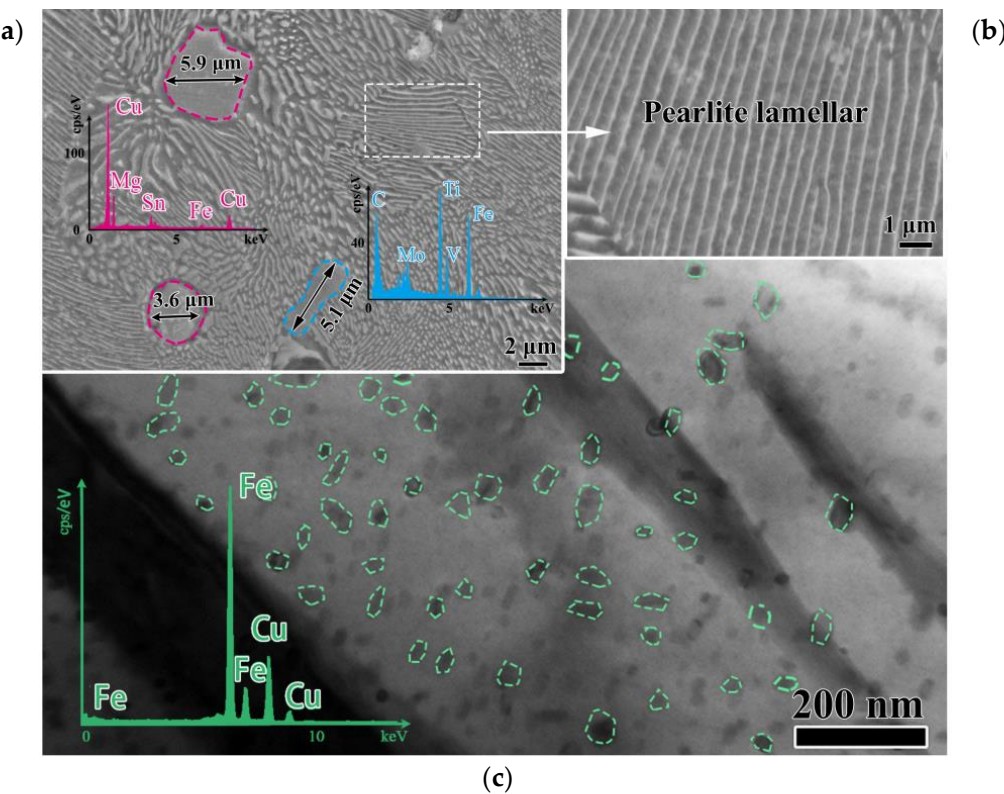

**Figure 5.** Microstructure of SHG: (**a**) SEM and EDS, (**b**) pearlite lamellar, (**c**) TEM.

### 3.2. Analysis and Comparison of the Wear Resistance of the Circular Spline

The 3D wear morphology of the three materials after friction and wear is shown in Figure 6. The morphology of the samples is marked with the corresponding length in the X, Y and Z axis directions. The SHF pattern shows narrow and short grooves with uneven wear surface depression, while the SHG shows an overall uniform wear pattern; in addition, the wear areas of SHF and SHG are concentrated. The overall wear pattern of 40Cr shows extremely uneven cutting grooves.

According to the vertical cross-section of the worn surface in Figure 7a, the maximum wear-mark depth of SHF, SHG and 40Cr is 3.8 µm, 4.2 µm and 7.2 µm, respectively. The grinding surface of SHF has a large fluctuation but a shallow grinding mark, while the grinding surface of SHG has a smoother fluctuation and a middle depth of the grinding mark. 40Cr shows the most fluctuating morphology with uneven grooves; there is 0.5 µm of grinding debris on the surface, and the depth of the grinding mark is the largest. The debris of 40Cr consists mainly of Fe, C and Cr elements, as shown in Figure 7b. The results are consistent with the former study of 3D morphology of wear.

The surface wear morphologies of the three materials are shown in Figure 8a,d,g. The width of the surface wear marks for SHF, SHG and 40Cr is 873.3 µm, 907.7 µm and 1090.3 µm, respectively. Compared with Figure 8b,e,h, it can be observed that the graphite nodules of SHF-20-50 are the most seriously elongated after dry friction, and all of them are elongated and deformed along the friction path. Relative to the high hardness of the matrix, graphite nodules are the first to exfoliate on the friction surface to form a lubrication layer, and graphite fully fills the volume of the worn groove to improve the wear resistance of the material.

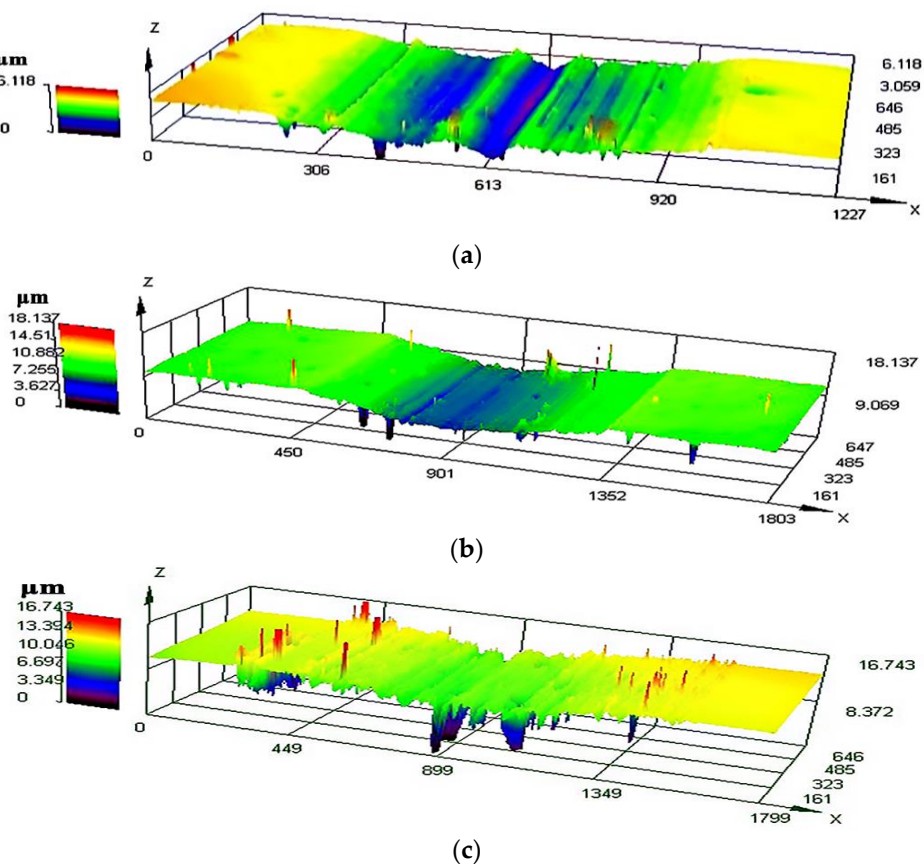

**Figure 6.** 3D morphology of the wear patterns: (**a**) SHF, (**b**) SHG, (**c**) 40Cr.

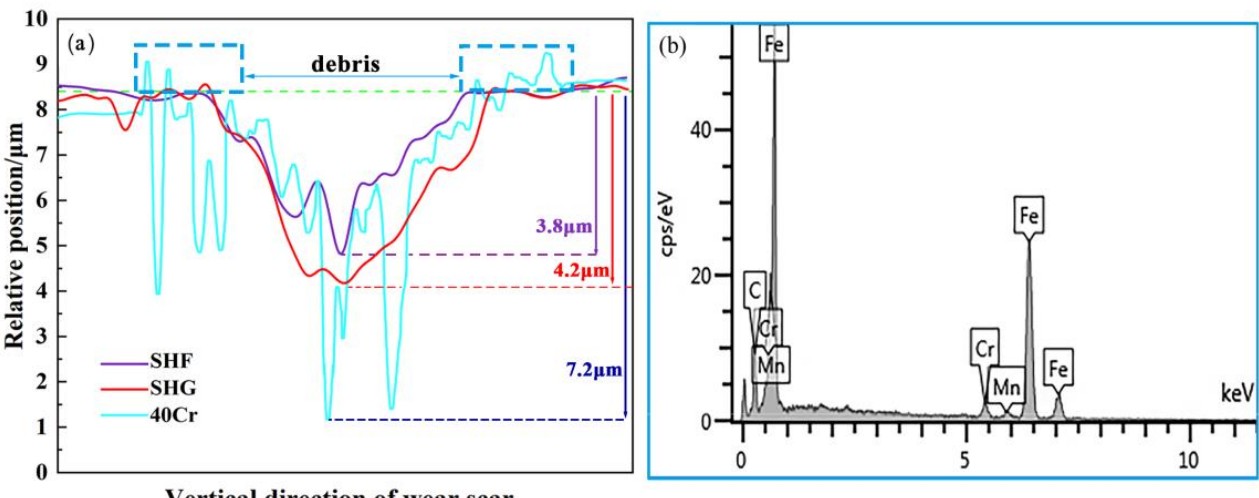

**Figure 7.** Morphology of the worn surface: (**a**) vertical cross-section, (**b**) the EDS of the debris.

Compared with Figure 8c,f,i, the surface of SHF is dominated by groove morphology, and at the same time, there is partial spalling and adhesion, which is mainly dominated by the abrasive wear mechanism. The normal force of the surface presses the abrasive particles into the friction surface, forming an indentation on the surface, and the tangential force pushes the abrasive particles forward, cutting the surface to produce a groove. The SHG circular spline is also dominated by the abrasive wear mechanism, and there is a lot of fatigue peeling on the surface, which is due to the increase in Cu content that greatly reduces the plasticity of ductile iron; in addition, the material on the friction surface is

peeled due to fatigue under the action of the cyclic contact force generated by the abrasive wear mechanism. The groove on the 40Cr surface is obviously shorter and thinner, the wear surface attaches to the groove in which there is a lot of strip adhesion along the friction direction, and the wear mechanism is mainly adhesive wear. 40Cr is more plastic than ductile iron, and the abrasive particles often slip on the surface, making only short grooves and pushing the material to the sides or in front. The failure of the adhesive junctions of plastic materials is dominated by plastic flow and occurs deep in the surface layer with large wear particles. However, when SHF and SHG were rubbed, the damage of the adhesive joints was mainly fatigue peeling, the damage depth was shallow, and the wear particles were small, which made them easy to fall off without accumulating and sticking to the surface.

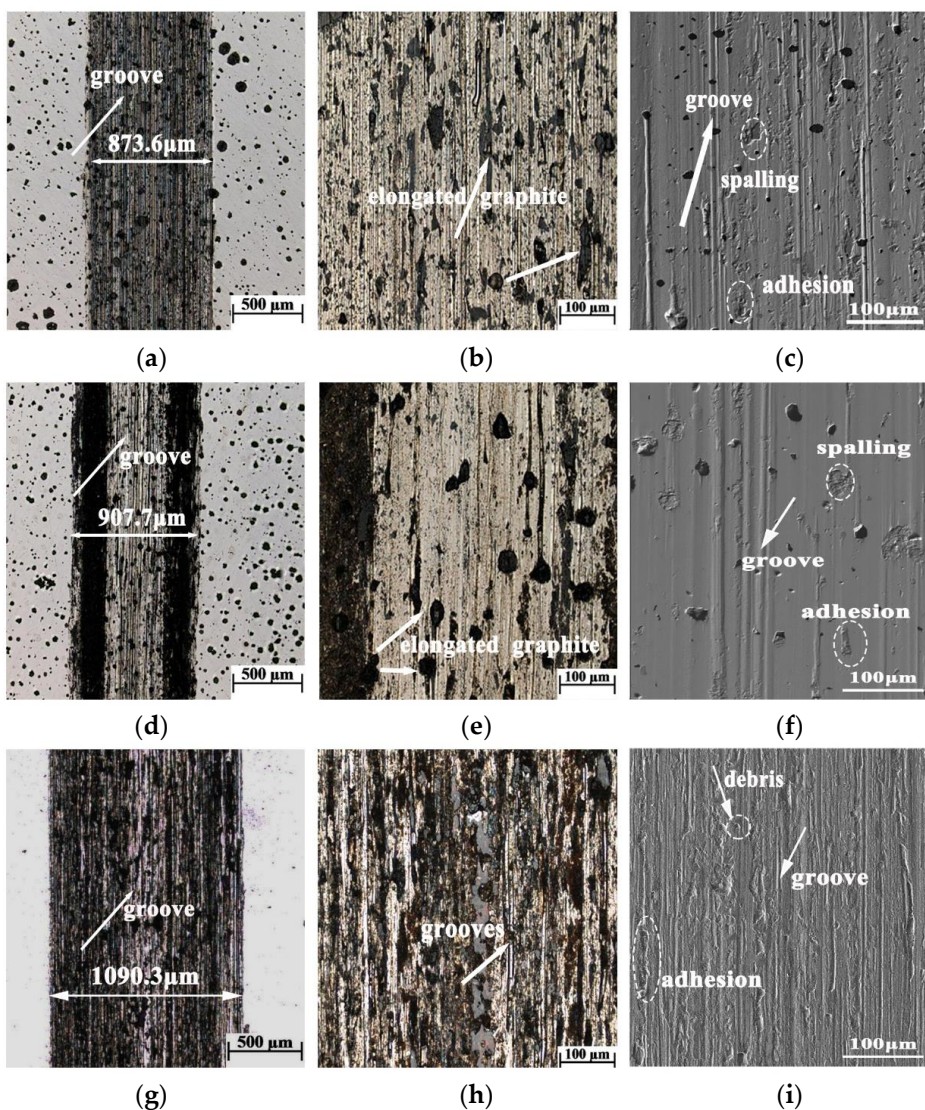

**Figure 8.** Wear microscopic morphology: (**a–c**) SHF, (**d–f**) SHG, (**g–i**) 40Cr.

The hardness values of the SHF, SHG and 40Cr circular splines are shown in Figure 9. The average Rockwell hardness of the SHF, SHG and 40Cr type circular splines is 27.1HRC, 28.6HRC and 30.4HRC, respectively. The average deviation of 40Cr, SHF and SHG is 0.8, 0.5 and 0.4, respectively. The hardness of 40Cr is the largest with a larger fluctuation, and the hardness of SHG is slightly higher than that of SHF with comparable hardness fluctuation at a lower level.

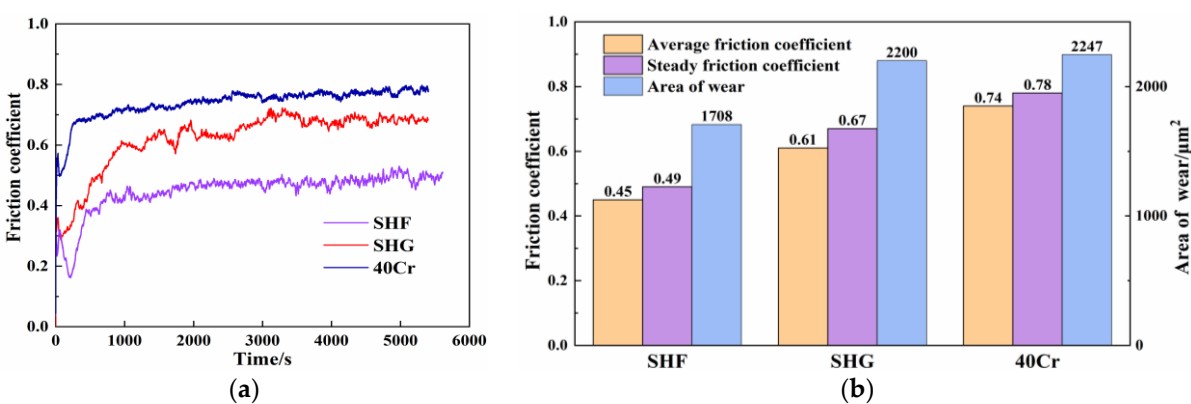

**Figure 9.** Rockwell hardness and its average standard deviation of the three materials.

The friction coefficient of the three circular splines is shown in Figure 10a. The average and steady friction coefficients are 0.45 and 0.49 for SHF, 0.61 and 0.67 for SHG and 0.74 and 0.78 for 40Cr. 40Cr entered the steady-state friction state in the shortest time followed by SHF, and both remained stable after 1500 s. SHG continued to maintain the steady-state friction state after 3000 s due to the existence of the graphite nodules; self-lubrication conditions were continuously established in the early stage of friction and wear. Therefore, it takes longer for ductile iron to establish a steady friction state than 40Cr.

**Figure 10.** Friction and wear process and final state: (**a**) dynamic change in the friction coefficient, (**b**) friction coefficient and the area of wear.

Taking the horizontal baseline in Figure 9 as the reference, the absolute wear area of the three materials was calculated with the integral method in the mathematical analysis of origin. The result is shown in Figure 10b; the wear areas are 1708 $\mu m^2$ for the SHF section, 2200 $\mu m^2$ for the SHG section and 2247 $\mu m^2$ for the 40Cr section.

## 4. Discussion

Compared with 40Cr, ductile iron has better wear resistance, which is due to the existence of graphite that provides a solid self-lubrication layer. In the process of continuous friction, it was peeled off to the surface of the wear mark under shear stress, and the matrix under the worn surface layer still had graphite to constantly supplement and participate in friction; this maintained a continuous lubricating layer, which enhanced the wear resistance of the matrix. SHF has a lower friction coefficient and area of wear than those of SHG. The difference between SHF and SHG can be explained from two aspects. On the one hand, the SHF graphite area fraction and diameter are larger. The larger graphite nodules on the matrix had a heavy splitting effect during the friction, were easier to peel and resulted in a longer lubrication effective distance. H.R. Abedi [33] studied the effect of nodule count on the sliding wear behavior of ductile iron and found that specimens with a high nodule count exhibited a lower wear rate than those that have a low nodule count at lower applied loads, while the wear resistance deteriorates with increasing nodule count at higher loads.

Therefore, more graphite nodules were extruded into pieces and flowed to the worn surface, further reducing the wear. On the other hand, it could be attributed to the difference in the carbide morphology. The carbide in tempered sorbite is distributed in the ferrite matrix in granular form. Carbide increases the wear resistance of the matrix without destroying the continuity of ferrite. However, in the lamellar pearlite, the carbide spacing is coarse, which destroys the continuity of the matrix [34]. When the hardness is similar, the fracture toughness is the key that significantly affects the wear resistance of brittle materials. Compared with tempered sorbite ductile iron, pearlite ductile cast iron has lower fracture toughness, and excessive Cu further reduces the plasticity of the matrix, so the surface is more prone to micro-cracks and block fatigue spalling [10].

## 5. Conclusions

In this paper, the dry friction and wear performance of ductile iron for two types of harmonic reducer at room temperature are studied and compared with 40Cr alloy steel for the circular spline. The relationship between the microstructure, hardness and wear mechanism was revealed, and the difference of wear resistance of the different materials was determined. The following conclusions are drawn:

1.  The microstructure of SHF is tempered sorbite, the microstructure of SHG is pearlite, and the pearlite lamellar spacing is 0.57 $\mu m$. The main enhanced precipitated phase in the former is (Ti, V, Mo)/C and in the latter is Cu/Mg; both are the second-phase particles, which are larger than 1 $\mu m$ in average length. The microstructure of 40Cr is tempered sorbite.

2.  Compared with 40Cr, ductile iron has better wear resistance, which is due to the existence of graphite that provides a solid self-lubrication layer. SHF has the lowest friction coefficient and wear followed by SHG, and the highest is 40Cr. The nodularity of the ductile iron of the two types of circular splines is well controlled, reaching more than 90%, and the obvious difference is mainly reflected in the area fraction and the diameter of the graphite nodules. The area fraction of the SHF graphite nodules is 8.58%, the area fraction of the SHG graphite nodules is 7.41%, and the average diameter of the graphite nodules is relatively thick, reaching 37.7 $\mu m$, while the average diameter of the SHG graphite nodules is 28.8 $\mu m$.

3.  The dominant wear mechanism of the SHF and SHG circular splines is abrasive wear, and the dominant wear mechanism of the 40Cr circular spline is adhesive wear. Hardness is not a key factor affecting the wear resistance of the materials. In this paper, the wear resistance is inversely proportional to the hardness, and the wear resistance

of the two types of ductile iron depends on the different microstructures and second-phase particles. The area fraction of the SHF graphite nodules is 1.17% higher than that of the SHG graphite nodules, and the average diameter of the graphite nodules is 30.9% larger than that of the SHG graphite nodules. Both the diameter and the area fraction of the SHF graphite are large, the effective lubrication distance is longer, and more graphite was extruded to the wear surface, further improving the wear resistance. The wear resistance of the tempered sorbite with (Ti, V, Mo)/C as the main precipitation is stronger than that of the pearlite with Cu/Mg precipitation phase.

**Author Contributions:** Writing-original draft: G.H.; Writing-review and editing: Y.G.; Data curation: T.W.; Funding acquisition: X.M.; Methodology: F.R.; Formal analysis: Z.S.; Supervision: C.Z. All authors have read and agreed to the published version of the manuscript.

**Funding:** This study was supported by the Students Research Training Program (SRTP) of University of Science and Technology Beijing (Project No. 2022031112).

**Data Availability Statement:** Not applicable.

**Conflicts of Interest:** The authors declare that they have no known competing financial interests or personal relationships that could have influenced the work reported in this paper.

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
