# Peer review of "Influence of Materials on Dry Friction and Wear Performance of Harmonic Reducer Circular Spline"

_metals, doi:10.3390/met13020378_

Round 1

Reviewer 1 Report

The review concerns a manuscript entitled "Influence of Materials on Dry Friction and Wear Performance of Harmonic Reducer Circular Spline"

Introduction
The authors should make a broader review of the literature, taking into account the results of the work of teams from different countries, including Europe.

Please do not use references to multiple items (e.g. [4-6], [11-16], [22-24]), but indicate what is characteristic of each of the cited items (what research was done, what results were obtained, how this relates to the issue od the manuscript).

Materials and Methods
Please provide more data on the research methodology, in particular the measurement data (working mode, magnification, etc.).

The Authors don't not write how many repetitions were performed, which is quite important information in this paragraph.

Results/Discussion
The figures are illegible. Please increase the font size and/or improve the contrast between the font and the background of images.

Figure 6. Has the surface been levelled? What do the peaks mean (in particular in Fig. 6 b and c)? - in my opinion, it may be measurement noise that should be filtered out ... or wear particles (?)

There is no discussion of the results referring to what other researchers have done in this regard.

Conclusion
This paragraph should be supplemented after appropriate additions have been made to previous paragraphs of the manuscript.

Reviewer 2 Report

Dear Authors,

Your article is interesting and gives current results. I have a few comments about the article:

1) you mention the use of three materials and their different microstructure, what were the heat treatment parameters? A heat treatment was performed or the starting microstructures were supplied by the material supplier. Please add information;

2) the legibility of the description, especially written in smaller font, is poor in Figures 4 and 5. Change the color of the description and remove the white background of the letters;

3) in Fig. 6 you document the 3D morphology of the wear. How many wear track measurements were taken and how many tracks were taken on each sample? Are the tracks you listed mean values? At what point in the track were the displayed morphologies documented?;

4) the track depths shown in Fig. 6 do not correspond to the values of the wear track profiles in Fig. 7. In Fig. 6 you can see the maximum depth values in the Z axis for the material SHF 6.118 µm, SHG 18.137 µm and 40Cr 16.743 µm. The profiles in Fig. 7 have depths for SHF 3.8 µm, SHG 4.2 µm and 40Cr 7.2 µm. Are the wear track profiles in Fig. 7 mean values from the analyzed 3D morphology (from a series of profiles) or is it a single profile? I value the depth of the profiles in Fig. 7 with the error of the standard deviation;

5) the wear track profiles in Fig. 7 are incorrectly displayed, they should be filtered with a suitable filter. Specifically, the information about the depth of the 40Cr material profile is completely misleading (you are measuring the dominant depression). By using a suitable filter (interpolation), you will get a profile shape similar to the SHG material (red curve) and this will also correlate with the values of the Z axis shown in Fig. 6.
